# Codesign and Feasibility Testing of a Tool to Evaluate Overweight and Obesity Apps

**DOI:** 10.3390/ijerph19095387

**Published:** 2022-04-28

**Authors:** Elisa Puigdomènech, Noemi Robles, Mariona Balfegó, Guillem Cuatrecasas, Alberto Zamora, Francesc Saigí-Rubió, Guillem Paluzié, Montserrat Moharra, Carme Carrion

**Affiliations:** 1Agència de Salut Pública de Barcelona (ASPB), 08023 Barcelona, Spain; empuigdo@aspb.cat; 2Red de Investigación en Cronicidad, Atención Primaria y Promoción de la Salud (RICAPPS), Barcelona, Spain; mcarrionr@uoc.edu; 3eHealth Lab Research Group, Universitat Oberta de Catalunya, 08035 Barcelona, Spain; 4eHealth Center, Universitat Oberta de Catalunya, 08018 Barcelona, Spain; 5Clínica Sagrada Família, Cuatrecasas-Peitx Endocrinologia i Nutricio Societat Limitada (CPEN SL) Servei d’Endocrinologia i Nutrició, 08022 Barcelona, Spain; mbalfego@cpen.cat (M.B.); gcuatrecasas@cpen.cat (G.C.); 6Corporació de Salut del Maresme i la Selva, Hospital de Blanes, 17300 Blanes, Spain; azamorac.zamora@gmail.com (A.Z.); gpaluzie@gmail.com (G.P.); 7Grup de Medicina Traslacional i Ciències de la Decisió, Universitat de Girona, 17003 Girona, Spain; 8Faculty of Health Sciences, Universitat Oberta de Catalunya (UOC), 08018 Barcelona, Spain; fsaigi@uoc.edu; 9Interdisciplinary Research Group on ICTs, Universitat Oberta de Catalunya (UOC), 08018 Barcelona, Spain; 10Agència de Qualitat i Avaluació Sanitàries de Catalunya (AQuAS), 08005 Barcelona, Spain; mmoharra@gencat.cat; 11Centro de Investigación Biomédica en Red en Epidemiología y Salud Pública, Barcelona, Spain; 12TransLab Research Group, Faculty of Health Sciences, Universitat de Girona (UdG), 17003 Girona, Spain

**Keywords:** codesign, participatory research, mHealth, overweight, obesity, pilot testing

## Abstract

Background: Digital health interventions and mobile technologies can help to reduce the rates of obesity and overweight conditions. Although weight management apps are widely used, they usually lack professional content and evaluation, so the quality of these apps cannot be guaranteed. The EVALAPPS project aims to design and validate a tool to assess the safety and effectiveness of health-related apps whose main goal is to manage and prevent obesity and overweight conditions. Objective: The aim of this paper is two-fold: (a) to co-create and codesign the EVALAPPS assessment tool and (b) to pilot its feasibility among overweight and obese individuals that use weight control apps. Methods: A mixed-methods approach was used. A multidisciplinary team (*n* = 12) participated in a co-creation workshop to provide proposals and inputs about the look and feel of the content, usability aspects, appearance, sections, and main features of the EVALAPPS tool. The tool was tested for its feasibility among 31 overweight and obese individuals, attending the CP Endocrinologia i Nutrició SL Clinic for the first time. Participants were asked to use a specific weight control app [Yazio (YAZIO GmbH, Erfurt, Germany), My FitnessPal (MyFitnessPal, Austin, TX, USA) or MyPlate (MyPlate, Santa Monica, CA, USA)] for two weeks and then evaluate them by using the EVALAPPS (EVALAPPS, David Ganyan, Barcelona, Spain) (June 2020, David Ganyan, Barcelona, Spain) tool. Seven participants were phone interviewed to gain more insight into the use of the EVALAPPS tool. Results: The co-creation workshop allowed conceptualizing the EVALAPPS tool. The feasibility study showed that all criteria from the Usability and Functionality dimensions had valid answers, while Reliability, Security, Privacy, and Health indicators were the dimensions with less valid answers. In all three apps, the dimension with the highest score was Usability/functionality, followed by app purpose. Clinical effectiveness and Development were the dimensions with the lowest scores in all three tested weight control apps. Conclusions: The participation of the multidisciplinary team and end-users in the conceptualization and testing of a tool to assess health apps was feasible and relevant for the usability of the tool.

## 1. Introduction

According to the WHO, since 1975, obesity has tripled: The number of adults worldwide that were overweight in 2016 was more than 1.9 billion, and from those, 650 million were obese [1]. Overweight conditions and obesity have adverse effects on individuals’ health, causing cardiovascular diseases, cancer, type 2 diabetes, depression, lower quality of life, respiratory disease, musculoskeletal disorders, and many others [2]. Factors influencing overweight conditions and obesity and treatments are complex and diverse [2]. Treatments for overweight individuals and those experiencing obesity should include multifactorial approaches, including diet, exercise plans, psychological therapy, and behavioral change strategies [3].

Health apps are quick, malleable, adaptable, convenient, and instructive tools that can play an important adjuvant role in the prevention and treatment of overweight and obesity. They can track physical activity (PA) and allow dietary patterns self-reporting, as well as provide guidance, recommendations, and tips to achieve healthier habits. Thus, digital health interventions and mobile technologies could help in reducing the rates of obesity and overweight conditions [3,4]. Two recent systematic reviews concluded that technology-based interventions can be effective for weight loss in overweight patients and those with obesity [5,6]. In 2016, NiKolau and Lean identified a total of 28,905 apps for obesity and weight management, most of them based on promoting healthy diets and exercise; the authors identified professional input in only 0.005% of these apps [7]. Therefore, although weight management apps are widely used, they usually lack professional content and evaluation. Tools and instruments to assess the quality of weight management apps are necessary to ensure a minimum quality standard for these apps.

The main objective of the EVALAPPS project is to design and validate a tool to assess the safety and effectiveness of health-related apps whose main goal is to manage and prevent overweight conditions and obesity. Previous phases of the project developed the content of the tool through (a) a systematic review to identify potential effectiveness criteria already used in former validation studies [8] and (b) a Delphi study to achieve consensus among a broad group of stakeholders on a comprehensive set of criteria to guide the development of the EVALAPPS assessment tool and to prioritize them [9]. While the systematic review aimed to consider the content of already available tools used for mobile health solutions assessment, the Delphi process, in which 31 stakeholders participated, generated a list of 53 criteria (from 114) that were considered essential for the EVALAPPS evaluation tool. These criteria were grouped in the following domains: security and privacy, usability, activity data (a health indicator), clinical effectiveness, and reliability dimensions. The development of the EVALAPPS tool and its assessment is framed in the World Health Organization Monitoring and Evaluating Digital Health Interventions A practical guide for conducting research and assessment [10]. The framework proposes monitoring goals, stages of evaluation, and an illustrative number of system users according to the stage of maturity of a specific digital health intervention. The current stage of maturity of the EVALAPPS tool is a prototype that, according to the WHO framework, corresponds to a functionality monitoring goal, a feasibility/usability stage of evaluation, and a range of systems users between 10 and 100. The aim of this paper is two-fold: (a) to co-create and codesign the EVALAPPS assessment tool, and (b) to pilot its feasibility among overweight and obese individuals that use weight control apps.

## 2. Materials and Methods

The current study consists of two stages aligned with both study aims: developing the format and sections of the EVALAPPS tool—the content was previously defined [8,9]—and pilot testing. In the tool development stage, a prototype of the EVALAPPS tool as an app was developed, and in the second stage, it was tested. A mixed method approach was used, including a workshop, objective app usage data, and individual interviews. No control group was included in the pilot testing since its main aim was to test its feasibility.

### 2.1. The Co-Creation Process: Development of the Format and Sections of the EVALAPPS Tool

A co-creation process was used to define and design the EVALAPPS tool. Participants in this process were members of the research team (including researchers, experts in health technology assessment, and healthcare professionals) and technology experts from the Open University of Catalonia. Contributors were asked about the design, sections, and key features of the tool. The content of the tool—criteria and dimensions to be used when assessing the quality of weight control apps—was defined through a systematic review [8] and a Delphi study [9]. Criterion (questions) included in the tool ranged from more plain user perspective questions to expert opinions. The co-creation process was framed in the Design Thinking process, which is an iterative and flexible process in which usually around 6–12 professionals (namely designers and users) identify, define, prototype, and test a specific service [11].

The co-creation process was done through a workshop in which participants provided proposals and inputs regarding the look and feel of the content, usability aspects, appearance, sections, and main features of the EVALAPPS app. The workshop was held on 22 November 2018. The co-creation workshop used different techniques and dynamics of creativity and gamification and was structured around the following challenges: characterization of the design and appearance and look and feel of the tool (access, dimensions to evaluate, structure, order, question types, navigation, completion, and look and feel) and the definition of the profiles of EVALAPPS users (single or multiple profiles). An expert on dynamization conducted the workshop, and three trained observers provided support when discussing in small groups. The workshop was audiotaped with prior consent from participants and transcribed systematically and verbatim. A theoretical thematic content analysis was performed based on the information gathered in the workshop. All participants signed a written informed consent form prior to their participation in the co-creation workshop. Appendix A shows the protocol of the co-creation workshop.

### 2.2. Pilot Testing: Feasibility Test of the EVALAPPS Tool

Overweight and obese individuals (Body Mass Index ≥ 25 kg/m^2^) attending for the first time the CP Endocrinologia i Nutrició SL (CPEN), an endocrinology unit in Barcelona, Spain, was offered to participate in the feasibility test. Inclusion criteria: participants who answered affirmatively to the following two questions: (a) Do you have a Smartphone? and (b) Do you think you have enough knowledge to install and use an app for weight and obesity control and the EVALAPPS assessment app? and agree to participate in the study. Those patients with little knowledge of the language of the apps to be tested (Spanish and English) and the EVALAPPS app (English, Catalan, and Spanish) and who could not guarantee participation were excluded.

Three nutritionists from CPEN SL oversaw patient recruitment at a rate of two patients per week per nutritionist. All patients who meet the inclusion criteria up to the expected 30 were invited to participate. The feasibility test study took place between October and December 2019. Each patient was randomly assigned to evaluate one of the three apps proposed by the CPEN center: Yazio, My Fitness Pal, and Myplate.

At the initial visit, after the patient was visited by the health professional and/or nutritionist, the recruiting (nutritionist) members of the research team instructed (10–15 min) the patients who agreed to participate in helping them become familiar with the app they would test for two weeks. Coinciding with a scheduled visit approximately two weeks after the initial visit and after a visit with the healthcare professional and/or nutritionist, the patient was instructed in the use of the EVALAPPS tool. Participants were then asked to assess the app for managing overweight and obesity using the EVALAPPS tool in the consultation or at home. Valid information was answers ‘Yes/no’ in dichotomous criteria and values ranging from 1 to 4 in Likert type items criteria. Non-valid answers were considered ‘I do not know’ and ‘I have not found the information.’ The tool included a ‘Reevaluation’ button that, if pressed, allowed the evaluator to stop the evaluation and continue with it later.

During March and April 2020, participants were invited to participate in an in-depth interview to gain more insights about the EVALAPPS tool after being used. Questions were about their difficulties and strengths when using EVALAPS and its strong and weak points. The interview included Likert-type items: participants were asked to rate the importance of dimensions assessed through a scale from 0 (not important at all) to 10 (very important). The information stored in the EVALAPPS tool was downloaded by the research team, and an analysis of the main key performance indicators was undertaken using STATA (v. 16, StataCorp, College Station, TX, USA). The information from the in-depth interview was grouped in aggregate terms and ideas. The information gathered in the Likert items was analysed using Excel (v17.0, Microsoft, Redmond, WA, USA).

## 3. Results

### 3.1. Co-Creation Workshop: Development of the Format and Sections of the EVALAPPS Tool

A total of 12 professionals participated in the co-creation workshop: 33.3% females. The professionals that participated in the workshop had expertise in patient management (general practitioners and endocrinologists; *n* = 3), health technology evaluation (*n* = 4), patient experience and empowerment (*n* = 2), and digital health interventions solutions development (*n* = 3).

#### 3.1.1. General Issues

The profile of the evaluator was one important general issue since the assessment could be different according to the profile of the evaluator, and even within the same profile, there may be different interests or basic knowledge. Additionally, there could be differences between professionals and users in relation to the objective of the evaluation, the content to be evaluated, and the basic knowledge of each profile


*“The approach is different, because for the patient it is a data collection and for the professional it is an evaluation tool.”*



*“There are indicators or dimensions that may be more typical of the health professional and others that may be more specific to the patient. For example, motivational issues are purely the patient’s, as the health professional cannot evaluate it. Instead, other aspects are more typical of the health professional.”*



*“Tracking, adherence, … are more users. Instead, efficacy can be shared and safety would be more health professional-like.”*



*“There are dimensions that are more typical of an ICT professional, or about how data is encrypted, etc. And a user or health professional cannot assess these criteria.”*



*“Me, as a patient, may be particularly concerned about data security or confidentiality. Instead, someone else may not have this concern.”*



*“On safety issues, depending on which diets are promoted, the patient and health professional may see it differently. For the patient it may be a motivational element, but instead, the health professional may consider that depending on what type of diets may put the patient’s health at risk.”*


However, for practical purposes, all participants agreed that a single tool should be generated, which could be used differently depending on the profile, so that each person could evaluate those aspects that interest him/her most and for which he/she has the enough knowledge.


*“If we make two profiles, we are already making two products. They may be pretty much the same questions for each other. To avoid complexity it is best to make a single application.”*


It was decided that evaluators should avoid answering questions on which they do not have sufficient knowledge, as this would detract from the results of the evaluation. Two strategies were proposed to differentiate between the different profiles: including a pre-questionnaire to discriminate the type of user, and consequently, the tool could guide the evaluation of certain dimensions. Another possible option was that the user could choose the dimensions he/she can evaluate (the ones that concern them most and on which they have an opinion). In this case, there would be no need to differentiate between profiles, but the content of the evaluation would vary according to each person.


*“Another possible option is to leave the choice at the discretion of each person so that everyone chooses the dimensions that they want to evaluate (the ones that concern them most and in which they have opinion). In this case, there would be no need to differentiate between profiles, but the content of the evaluation would vary by person.”*



*“That each person can choose, as a user, the criteria that he/she wants to value of the app, because they are the ones that he or she cares about the most. More than creating multiple profiles.”*



*“Two profiles will always be there, and perhaps the decision is if you ask at the beginning when you create the profile to discriminate, but this can be very aggressive for users who make use of the application because they already have the feeling that “they will study them.”*



*“In any case, entry cannot be aggressive and differentiating between patients and health professionals can create a barrier. The question should go depending on the use that will be made of the application (as a patient, to recommend it to patients, …)”*


#### 3.1.2. Content of the Application

Table 1 shows the main themes related with the content of the application. For each theme, main ideas are described and verbatims that support the idea are offered.

#### 3.1.3. Dimension Weights

Workshop participants were also asked to weigh the importance of each dimension in the evaluation process of weight control apps. Participants reported higher weights for the following dimensions: usability (18%), quality of information and content (15%), clinical effectiveness (13%), and security and privacy (12%). The rest were weighted as ten or lower, with health indicators being the dimension with the least weight (3%). A recurrent idea regarding the weight of the dimensions could be related to the evaluator’s profile: for example, more weight could be given to clinical dimensions if the evaluator was a health care professional.

#### 3.1.4. Mobile Phone EVALAPPS Tool

The content of the tool was defined in previous studies of the project [8,9], and the sections and appearance were defined in the co-creation session. Both an expert app developer and an expert app designer helped the research team with both the technological development and the design of the app, respectively. Both included improvements proposed by the research team through an iterative process that lasted four months.

The tool is a user-friendly app available on IOS and Android systems that generates data related to the evaluation of weight control apps. Before undertaking the evaluation, the EVALAPPS tool provided brief information on the objective and content of the evaluation in video format. The evaluation is proposed as a list of the dimensions to be evaluated, with a follow-up of the level of response performed, so that, if the evaluation is not completed, the tool can be subsequently entered, and the pending items continue to be evaluated. Namely, it can record information on 39 criteria grouped in seven dimensions: Purpose (three criteria), Development (three criteria), Reliability (five criteria), User functionality (7 criteria), Health indicators (13 criteria), Clinical effectiveness (five criteria) and Security (seven criteria). Each criterion could be rated by the evaluator as a dichotomous (yes/no with a smile or sad face) or as a three-grade scale. The options ‘I do not have enough information’ or ‘I could not find the information’ could also be marked by the evaluator (see Figure 1). It includes pop-up screens that provide information on each criterion and dimension being evaluated. To avoid monotony, each dimension was set in a specific color, both yes/no questions and Lickert-item questions were used, and ‘smiley’ and ‘sad’ faces were included. Once the assessment is performed, the EVALAPSS app provides a report with an overall score and a score disaggregated by dimensions. The app can be downloaded in the following link: https://play.google.com/store/apps/details?id=com.deltadev.EvalApps&gl=ES (accessed on 20 February 2022).

### 3.2. Pilot Testing: Feasibility Testing

A total of 31 patients participated in the feasibility testing. Table 2 shows sample characteristics. Approximately 75% were women and the most frequent age groups were 18–25 y.o. and 36–45 y.o. (29.0% each). More than 50% of the sample used an iOS Smartphone. Most evaluators used Spanish as the most frequent language when using the EVALAPPS app.

Although participation was offered to 31 individuals, a total of 24 registers were obtained (22.6% of registers were missed, *n* = 7). Appendix A shows missing values for each criterion and the percentage of valid information gathered through the pilot testing. As we move forward through the criteria to be evaluated, the number of lost values increases; except for the first criteria, all the rest presented at least one missing value. Criterion 28 further shows that the number of missing values exceeds ten (41.7% of missed answers). The maximum missing values in specific criteria were 13 (54.2% of lost answers) and are embedded in the following dimensions: Clinical effectiveness (all criteria), Security (all criteria, except one with 12 missing), and two criteria under Health Indicators dimension. All criteria from the Usability and Functionality dimension had valid answers. Reliability, Security and Privacy, and health indicators were the dimensions with less valid answers.

Table 3 shows the global punctuation of each tested app as well as for each specific dimension. The Yazio App showed the highest total score, with a mean punctuation of 44.7 (SD: 23.9), followed by MyFitnessPal (mean total score: 37.8 (SD: 17.6)), and MyPlate (mean total score: 31.3 (SD: 17.6). In all three apps, the dimension with a highest score was Usability/functionality, followed by app Purpose. Clinical effectiveness and Development were the dimensions with the lowest scores in all three apps.

#### Feasibility In-Depth Interviews

A total of seven telephone-based interviews were undertaken to gain insights into how users used the EVALAPPS app. Interviews were done with those individuals who consented to be called by the research team to gain insight into their use of the EVALAPPS tool. One of the interviews gathered only information on the app being evaluated since the person explained that she was not able to use the EVALAPPS app, so the following results are based on six interviews, two men and four women. Three participants evaluated MyFitnessPal, two MyPlate, and one Yazio. Only one participant confirmed he had problems with the EVALAPPS app installation on his Mobile phone; 4 had some problems uploading data, and the nutritionist had to help them. Participants reported that the evaluation of an app (by using the EVALAPPS app) lasted between 10 and 20 min; except for two participants, the rest declared it was the correct time. Four participants understood what the research team asked them to do, and two declared it was complicated. Most of the participants declared that some questions of the evaluation were difficult to answer according to their expertise or could not find the information on the app being evaluated. The answering system was well understood (dichotomous/Likert items) by the participants. Three participants declared they had some problems understanding the ‘Revaluation’ button since they assumed that they had not performed the evaluation and that data had not been stored.

## 4. Discussion

The study aimed to provide the appearance and testing of the EVALAPPS tool to assess weight control apps through a co-creation process with a research team and technology experts. The main findings of the co-creation process included: the format of the tool as a single app that could be used differently depending on the profile of the evaluator. Namely, the tool should also provide information on the evaluation process and the dimension and criteria to be evaluated. A combination of yes/no questions and Likert-item questions should be used with strategies to avoid monotony (faces and the use of color scales for each dimension). The co-creation process defined and designed the EVALAPPS tool, while the results of the pilot study in which the first version of the tool was tested introduced changes and improvements. On the other, the first version of the EVALAPSS app was piloted among three weight control apps for overweight and obese individuals. All criteria from the Usability and Functionality dimensions had valid answers, while Reliability, Security and Privacy, and Health indicators were the dimensions with less valid answers. In all three apps, the dimension with the highest score was Usability/functionality, followed by app purpose. Clinical effectiveness and Development were the dimensions with the lowest scores in all three apps. The final tool is intended to serve as an assessment tool to help end-users assess the quality of weight management apps. The results can also be useful for future APP developers.

Several organizations have developed assessment initiatives to evaluate health apps in general or by themes. For instance, the World Health Organization [12], the United Kingdom National Health System [13], the Royal College of Physicians [14] or Stoyanov, and colleagues [15] have developed assessment tools and checklists for general health apps assessment (MARS), which include the following dimensions to be assessed: engagement, functionality, aesthetics, information, and subjective quality on 5-point scales. Several authors are using this checklist to assess different types of health apps, such as drug information and drug-drug interactions apps [16], self-management diabetes apps [17], or oral hygiene apps [18]. Bardus and colleagues used the MARS tool to assess the quality of six weight management apps (My Diet Coach, SparkPeople, Lark, MyFitnessPal, MyPlate, and My Diet Diary) among 36 Lebanon university workers. They reported that the highest proportion of missing items was from three information dimensions (three items: the credibility of source: visual information, and quantity of information: 22%) and one in engagement (customization). All assessed apps scored above 50% in the MARS score, and the highest scored dimensions were functionality, aesthetics, and information [19].

Specifically, for the assessment of nutrition apps, in 2017, DiFilippo et al. [20] developed the Nutrition App Quality Evaluation tool to evaluate nutrition-related apps. The tool includes 25 items grouped in five blocks: app purpose, behavior change, knowledge and skill development, app functionality, adding information about the app, and the user. It also includes five additional items to assess app appropriateness for different age groups and four additional items to assess apps for specific audiences [20]. All these evaluation tools had been reviewed in previous stages of this project. In 2019 research team reviewed all dimensions and criteria from these tools and grouped them into several thematic dimensions. After, through a Delphi process with main stakeholders, domains and criteria were prioritized. Consequently, the present tool has been developed by undertaking a review of the content of other available tools and by prioritizing the items that should be included.

### Strengths and Limitations

The co-creation process and the feasibility testing can pose a great strength in the future use of the EVALAPPS tool since it can help to produce a viable user-friendly tool that is practical and desirable for the target audience. The stage of the project has not allowed the research team to validate the tool but to test its usability and feasibility. Due to practical and budgetary reasons, no end-users were invited to the co-creation process of the tool. However, the posterior pilot testing was done among end-users, and they were offered the opportunity to provide feedback and propose changes to the tool. Specifically, seven interviews were undertaken to gain insights into the end-users when using the first version of the tool. Their feedback, whenever possible, will be included in the next versions of the tool. Pilot testing was done in overweight and obese individuals, and results could differ in the case of the general population using the app; however, weight control apps are widely used by the general population (including obese and overweight individuals) [21]. The pragmatic sample size in this study might be considered somewhat low; however, it is worth noting that this is a pilot study. Self-selection bias might have occurred among respondents since participants in the pilot study were recruited in a Clinic in Barcelona in which two professionals were researchers of the project and participated on a voluntary basis. Most of the participants were women from the following group ages: 18–25 and 35–46. This cannot guarantee the external validity of the results. Participants used both English versions of the apps or adapted versions of them for the Spanish-speaking users. Ratings might have differed if users had been the specific target audience; some of the apps used units of measure and types of foods used in the Anglo-Saxon culture. Ratings might have been different if users had tested premium versions of the apps, which have more functionalities instead of the free version. The research team asked participants to test the weight control apps for two weeks. However, we were not able to ensure that this instruction was done by the participants.

One of the strengths of the study is that data was stored in the Evalapps app, and consequently, missings were only related to those in which the participant decided not to respond. In-depth interviews were undertaken in a subsample of participants to gain insight after the use of the Evalapps App, which allowed the research team to check if the tool was feasible, understandable, and if minor modifications should be introduced.

## 5. Conclusions

This study shows the implementation of a co-creation methodology to create a tool to evaluate apps for weight control as well as a posterior validity testing among obese and overweight individuals. The present study can serve as an example to other researchers when developing co-creation processes in evaluation apps and fill the gap on this topic in the literature when framed in the design thinking process. The co-creation process allowed both the research team and end-users to provide feedback on the tool so that it could be tested by the general public. Further testing in the general population has been undergone with the final tool, which includes proposed adjustments and adaptations that were considered necessary to be included with the feedback provided by the feasibility study. For instance, in the final tool, the criterion to be answered appears randomly, and some language adaptations have been made (EvalApps—Aplicaciones en Google Play). The final tool could be adapted not only to assess weight control apps but could be used in other conditions, such as for the assessment of mental health apps.

## Figures and Tables

**Figure 1 ijerph-19-05387-f001:**
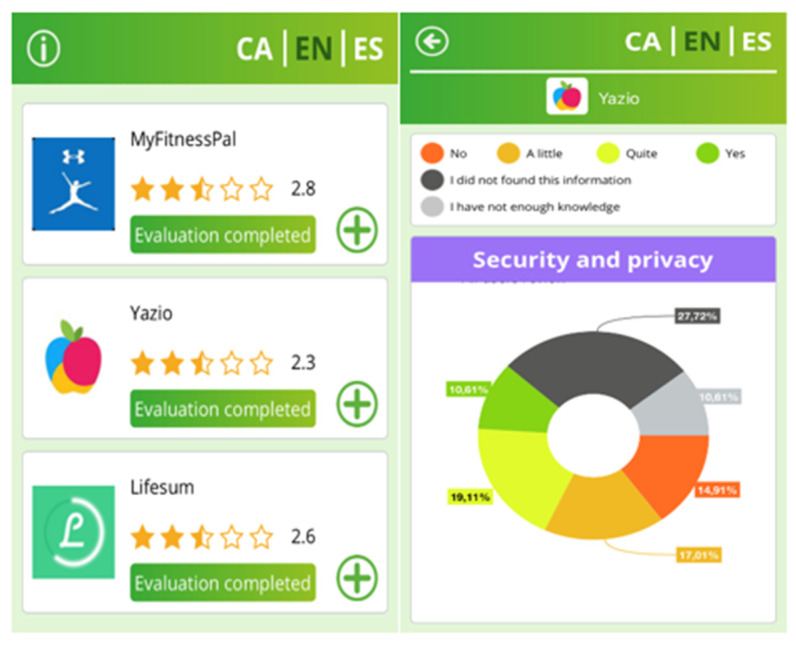
Screenshots of the Evalapps app.

**Table 1 ijerph-19-05387-t001:** Main themes and ideas related with the content of the application.

**Access to the Application: Log-In Free Access vs. Login Access**
Login free access:
✓ Prevents the request of personal data (username, mail) which can be a barrier to unsubscribe ✓ Can be difficult to control multiple evaluations from the same user in case he or she uses different devices.
Login access:
✓ Allows the identification of each user and the collection of useful information for the exploitation of the evaluation results. ✓ Must indicate very clearly authorship and objectives and ensure that it is not associated with any advertising. ✓ It is highly recommended that the app remembers the password automatically if prompted by the user.
**Evaluator’s profile**
Gathering the evaluator’s profile (background, exertise, …) can be used to:
(a) To assign the dimensions to be evaluated
✓ If the evaluation is conducted through certain dimensions according to evaluator’s profile, it is necessary to collect certain information about the evaluator. An introductory question can be included regarding the purpose of the evaluation. An example could be: What is the purpose of the evaluation? (the evaluator can check more than one option): As a health professional/As a user/As an ICT professional/As a design professional/Other use. ✓ If the evaluator decides which criteria and dimensions he or she will evaluate regardless of his/her background the information will not be necessary.
(b) To weight the user-provided responses, depending on the profile.
✓ Evaluator information can be used to weight responses based on his/her profile, for example by giving more weight to the responses of healthcare professionals in clinical dimensions.
(c) To use the results of the evaluation for research purposes.
✓ The analysis of results can be done according to the following variables: sex, age groups, and evaluator’s profile (user, health professional,…)
*“To have the minimum of demographic data to be able to segment the information we simply ask if it is male or female, the age it is, and we have a set of profiles (medical professional, user, technologist…) that can mark more than one, and in theory we do not ask for anything more”*
✓ This exploitation will be indispensable during the pilot and should be valued for the commercial version. For the pilot it is commented that some of this information can be obtained outside the application, linking the information of each user to their medical history.
(d) *“Depending on the studies you want to do now you have to differentiate the data that is asked to do the analysis of the results of the pilot, you have to evaluate what is requested in the app and what information is extracted outside the app (for example, the medical record). Because if that information is not going to be asked later in the app, it does not make sense to put it in the pilot and it is easier to extract it by other means outside the app”*
✓ Beyond the pilot, the need to ask the users of the application for sociodemographic information should be valued. This decision should consider the advantages and disadvantages of collecting information about users: on the one hand it allows information to be available to carry out investigations.
*“This is a clinical trial. And you must be able to have the information of the people who answer you in order to analyze the results you have obtained. For example, you might want to know if 40-year-old women answer differently than 60-year-olds, or if people with different degrees of obesity give more weight to some questions than others. And eventually you must be able to do it. So you have to design the tool thinking that this information you can have it… An extension of this is that this page is put at the end of everything so that it does not condition the entry… It’s not minimizing tools that you can then use in the future”*
✓ On the other hand, it can be a barrier to the use of the application.
*“We have left the premise that we did not want to ask the user for any information. Because it’s an entry barrier and we didn’t see what services we could offer to justify this barrier of entry.”*
✓ Finally, the discussion arises regarding when the collection of sociodemographic data should take place. Thus, while for some of the workshop participants these questions should be asked at the end of the evaluation, others argue that it should be done at first to minimize the risk of not collecting such information in the event that the person leaves the application before completing the evaluation and does not answer the profile questions.
Selection of the APP to be evaluated
✓ Research will select the weight control apps to be evaluated and by which participant. Technical aspects such as the mobile operating system version of the person performing the evaluation (iOS or Android) should be considered.
*“For the pilot it will be a closed list. You also have to see which app is recommended to each person, because there are apps that are on Android and others in IOS, so you have to differentiate depending on the device… ”*
✓ The selection of apps to be evaluated after the pilot is finished can be displayed in a carousel format, differentiating between recommended apps (e.g., apps with more than n downloads) and other apps. Moreover, the evaluation tool could identify the applications that the user has installed on their mobile phone and incorporate them into the list of applications to evaluate.
✓ *“You can theoretically know which apps are installed on a mobile. If you can tell which apps you have installed on your mobile, the quit is that the app you have to evaluate is one of the ones you have installed on your mobile. Have a carousel of the apps that you have installed on your mobile and select the one you want to evaluate.”*
✓ The debate also arises as to whether any application should be able to be chosen to be evaluated, or whether the applications to be evaluated should be pre-determined from the application itself. Leaving the possibility open carries the risk of incorporating applications that have nothing to do with health (e.g., pokemon go), but it was also valued that the objective of the application is to empower the user and that it should be given the possibility to upload any application so that the user can check if they are using an application that does not meet the basic requirements set by the evaluation.
Content of the evaluation
✓ Brief information on the objective and content of the evaluation should be provided to the evaluator. ✓ The evaluation is proposed as a list of the dimensions to be evaluated, with a follow-up of the level of response performed, so that, if the evaluation is not completed, the tool can be subsequently entered, and the pending items continue to be evaluated.
*“You’re entering information. It’s not the concept of “I answer a survey” but it’s an application in which you enter information over time. And it’s being kept, and if you do it on the bus, you’re halfway there. and after 2 days you get a notification that tells you “just completed”*
✓ It is proposed to undertake the assessment through a combination of yes/no questions and Lickert-scale questions. To avoid monotony, different strategies are proposed:
-change the way scales are presented, combining different types of icons (stars, faces, etc.);
-use different color scales for each dimension;
-use icons related to the content of the dimension to be evaluated;
-when a dimension evaluation is complete, an intermediate screen appears with a chart collecting the scores collected in that dimension throughout the evaluation process.
✓ The number of criteria to be evaluated is very high, so different options are raised to facilitate the answer:
-Present questions randomly, to prevent questions presented at the end from being systematically evaluated automatically;
-Start with the easiest questions to answer, and increase complexity as you progress through the assessment;
-Start with the most relevant questions to make the assessment, to ensure that you get the most answers to these questions, and then ask the rest.
✓ In relation to navigation options, depending on the number of criteria to be evaluated different options are proposed:
-Distribution of dimensions by tabs;
-List followed by criteria and scroll navigation;
-It was also suggested to incorporate information (using a pop-up screen) on the criteria to be evaluated, next to each question.
✓ The re-evaluation of the app can be a possibility, after a certain period after being evaluated. To do this, the tool would generate an automatic message between 15 days and 1 month after the user has evaluated the application so that it can be re-evaluated.
Report
✓ Once the evaluation is carried out, it is proposed to provide a report with different levels of information:
-Overall score;
-Score disaggregated by dimensions;
-Comparison between user and median score obtained during the evaluation process (e.g., using spider charts).
✓ The final evaluation report could be conceived as an incentive to the user (to be taken to something) and indicates the need to offer some other type of incentive, once completed—for example, a report with the profile of the most recommended applications, or a detailed application report…
*“In the end a report, with your assessment, the evaluation of other people and, below, an app recommender”*
Gamification
✓ The commercial version of the application, incentive for the use of the application, thorough gamification techniques
*“We have to find a way to gamify, to reward the patient… “*
*“The whole issue of compensation, the rewards to the user… they have to be on the platform and they can be built over time.”*
✓ The main incentive of such an application could be the provision of information about the evaluation carried out by other people
*“You’d have to see the evaluation of the others, the recommendations of the professionals and many more things”*
*“At the end of each page we gamify a little by teaching what people have answered to those questions, so that it is compared to others”*

**Table 2 ijerph-19-05387-t002:** Sociodemographic and technological characteristics of the EVALAPPS pilot testing.

	*n*	(%)
Gender		
Female	23	(74.2)
Male	8	(25.8)
Age group		
18–25	9	(29.0)
26–35	5	(16.0)
36–45	9	(29.0)
46–55	2	(6.5)
56–65	4	(13.0)
>65	2	(6.5)
Operating system		
Android	14	(45.2)
iOS	17	(54.8)
App evaluated		
MyFitnessPal	10	(41.7)
Yazio	6	(25.0)
MyPlate	8	(33.3)
Language used when using EVALAPSS tool		
Catalan	3	(12.5)
Spanish	20	(83.3)
English	1	(4.2)

**Table 3 ijerph-19-05387-t003:** Dimension and total score for each evaluated app.

	Yazio (*n* = 6)		MyPlate (*n* = 8)		MyFitnessPal (*n* = 10)	
	Mean (SD)	Min–Max	Mean (SD)	Min–Max	Mean (SD)	Min–Max
App Purpose	8.5 (3.4)	4–12	7.3 (3.6)	3–12	9.1 (2.7)	2–12
Development	1 (1.1)	0–2	0.7 (0.5)	0–1	1 (0.8)	0–2
Reliability	4.6 (4.6)	0–11	2.9 (2.1)	0–6	4.6 (3.3)	0–11
Usability	17.8 (10.9)	0–27	13.8 (9.7)	0–27	15 (10.1)	0–27
Health indicators	6.3 (3.9)	0–11	4.8 (4.3)	0–10	4.2 (4.2)	0–10
Clinical effectiveness	0.8 (1.16)	0–3	0.1 (0.3)	0–1	1 (1.6)	0–4
Security/Privacy	5.5 (5.4)	0–12	1.5 (2.9)	0–8	2.9 (3.6)	0–9
Total	44.6 (23.9)	4–65	31.3 (17.6)	3–55	37.8 (17.6)	10–62

SD: Standard Deviation; Min: Minimum; Max: Maximum.

## Data Availability

The data presented in this study are available on request from the corresponding author.

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
