# Peer review of "Codesign and Feasibility Testing of a Tool to Evaluate Overweight and Obesity Apps"

_ijerph, 2022, doi:10.3390/ijerph19095387_

Round 1

Reviewer 1 Report

This research examines the co-design and proof of concept of a tool for assessing overweight and obesity apps. The paper has two objectives first to co-design EVALAPPS assessment tool and second to test its feasibility in overweight and obese individuals using weight management apps.

In Introduction, current general tools for testing apps and general software, especially in the mHelath area, could be dealt with in more detail, since these methods are certainly transferrable or have been transferred to the EVALAPPS project.

Second paragraph materials and methods is split in two sections format and sections and feasibility test of the EVALAPPS tool, both seconds describe in detail what steps were taken what I am missing here is a general description of the methods on which these investigations are based the authors could go into more detail here in both sections.

The results of the two tests are discussed in detail. I lose track of the large tables with the many statements "General issues" and tables "Content of the Application", in my opinion it would be better to have the results of both investigations (Section 3.1 and 3.2) better summarize and evaluate. Do all statements have to appear here? Wouldn't it be better to look closer into the core statements and examine them? At least it would be good at the End of paragraph 3 to give a short summary with an evaluation of the results maybe a short table in discussion paragraph like an overview of the core results of investigation.

Conclusion could be divided into conclusion and outlook with more thoughts on future improvements and use for medical apps with concrete examples.

Author Response

We have provided the answers in a Doc file (attached)

Reviewer 2 Report

This paper aims to design an instrument called EVALAPPS and "pilot its feasibility among overweight and obese individuals that use weight control apps". Overall, the topic of the paper is relevant and interesting. However, the paper suffers from a number of weaknesses, which need to be addressed prior to the paper could be considered for publication.

1) The abstract is heavy, long, and somewhat hard to follow. The authors should consider a major rewrite in terms of making it more readable and accessible. Aside from other issues, the conclusion is very generic and does not highlight the key finding(s) of the authors' study. I am also not sure that the next steps of the project need to be highlighted in this part (the body of the paper seems to be fully adequate).

2) The methodology of the paper is rather unconvincing. I see tho major issues here. First, the reader gets virtually nothing regarding data analysis of the qualitative data from the workshop. The statement "The transcription was used as the primary source for the analysis" (line 114) is not enough. If, for example, thematic analysis was used, please explain in detail. Second, the same basically applies to the quantitative part. Also, clarify "The assignment 131 of the app to be assessed was done blindly by the nutritionist." (line 131). I am not sure what this means. Were the nutritionist really blinded? Who then decided about what user gets which app? Please elaborate... Finally, in-depth interviewing does not really resonate with Likert-type scaling (line 145-149). This should be better explained.

3) There are some logical inconsistencies across the paper, showing problematic coherence of the respective parts. For example, it is argued: "Content of the tool was defined in previous studies of the project [8,9]" (line 225), and, at the same time, "The co-creation process defined and designed the EVALAPPS tool" (line 298). Similar inconsistencies are very confusing for the reader.

4) In connection with the previous issues, the qualitative part seems to be solely a "dump" of the notes/data. There is virtually no (sophisticated) qualitative analysis presented, and that fact results in a very challenging reading experience. At times, it is not clear whether a specific statement is based on the data, or if it is the authors' statement, e.g. "Evaluator’s should avoid answering questions on which they do not have sufficient knowledge, as this would detract from the results of the evaluation." (line 189). It should be made clear that this was derived from the data by stating, e.g. "The workshop participants also held that ..." BTW What is (a,b,c) in the second heading of Table 1? 

5) To be honest, I didn't really get why you chose this particular sample of 12 professionals: "3 computer engineers, 3 medical doctors, 2 biologists, 1 chemist, 1 psychologist, 1 philologist and 1 statistician." (line 157). Can you clarify? This is very important in terms of the validity of your study.

6) Some of the questions (Table 4) are kind of weird and cast strong doubts regarding the contribution of the study. I don't really understand the point of questions such as "Is the App available in iOS and Android?" or "Do App developers generate trust / credibility?". This might be due to a problematic translation, though. Nevertheless, it seems that you mix a plain user perspective ("Does the App have a friendly and intuitive interface?") and an expert opinion (e.g. "Does the App send any alert in case it detects risky behaviors? (for example, very strict diets)"). Why? And to what extent you consider this being a valid approach? Not surprisingly, this fact seems to be reflected in the number of missing answers. BTW. Why the percentages in Table 4 don't sum to 100 %?

7) "Due to practical and budgetary reasons, no end-users were invited to the cocreation process of the tool" (line 339). Is then possible to call your approach co-creation and co-design at all? This severe limitation should be very clearly tackled right from the beginning of the manuscript because otherwise the reader is just lost (see the above comment).

8) The paper should clearly discuss the additional limitations of this research. For example, it should be highlighted that the point of the research wasn't to validate the instrument, and what that fact implies. To be brutally honest, after reading the manuscript I am not convinced that the proposed approach is "feasible" at all.

Minor issues and other observations:
- Please distinguish between Likert-type items vs. Likert scale (the sum). Also, you should clarify how you counted the total scores in Table 5 (by simply averaging the items?). In addition, check the consistency of the categories (e.g. App purpose - Table 4 vs. Purpose - Table 5).
- "Adhesion" (line 172) -- do you mean adherence?
- "generates data" (line 230) -- collects data?
- "Evalapps send-;" (line 248) -- ??
- Min-Ma (Table 5) -- "Min-Max"
- Etc. -- a number of similar minor issues could be found

Author Response

(The authors gave the same response as above.)

Reviewer 3 Report

The article is well written!

There are too many tables; perhaps Table 4 can be moved to Appendix.
Table 3: why BMI is not reported?

Is there a reason to categorise Age and report in groups?

It’s a* design and pilot testing. The paper might be more interesting if presented after the evaluation on the general population that uses the weight control apps.

Author Response

(The authors gave the same response as above.)

Round 2

Reviewer 2 Report

I appreciate the authors put some moderate effort into improving their manuscript in order to increase its readability and soundness. The majority of my comments were addressed. In general, the manuscript, including the abstract, now reads better. While I still consider the manuscript being somewhat immature in terms of publication standards in respected journals, it is basically up to the editor to judge the quality according to the standards expected in IJERPH. 

The areas which might be improved some more:

1) What type of thematic analysis was used? Include a literature reference. How the process exactly looked like?

2) The authors' reaction to comment no. 7 (the nature of involving end-users) should be somehow incorporated into the manuscript, at least by briefly mentioning that fact. 

3) "The development of the EVALAPPS tool and its assessment is framed in the World Health Organization Monitoring and Evaluating Digital Health Interventions A 83 practical guide to conducting research and assessment." (line 82-83): include a reference, and polish the sentence/citation format a bit.

Note:

"The total score was obtained by adding the value of all the items. Table 4 (previously 5) shows the mean, standard deviation and maximum and minimum values of the scores provided by the subjects who evaluated each app. " --> there is no Table 4 in the current version of the manuscript (which might be a good decision, though).

Author Response

Response to Reviewer 2 requirements are attached in the following document 'Response to reviewer 2b'
